# Can the Sci-Tech Innovation Increase the China’s Green Brands Value?—Evidence from Threshold Effect and Spatial Dubin Model

**DOI:** 10.3390/e25020290

**Published:** 2023-02-03

**Authors:** Xiaofei Zhang, Yang Xiao, Linyu Wang

**Affiliations:** 1School of Economics and Management, Fuzhou University, Fuzhou 350108, China; 2School of Economics, Zhejiang University of Finance and Economics, Hangzhou 310018, China

**Keywords:** green brand value, innovation efficiency, innovation value chain, intellectual property protection, negative entropy flow, spatial Dubin model, panel threshold model

## Abstract

Based on the perspective of the innovation value chain, sci-tech innovation is divided into two stages: R&D and achievement transformation. This paper uses panel data from 25 provinces in China as the sample. We utilize a two-way fixed effect model, spatial Dubin model, and panel threshold model to discuss the impact of two-stage innovation efficiency on the value of the green brand, the spatial effect of this impact, and the threshold role of intellectual property protection in the process. The results indicate that: (1) the two stages of innovation efficiency have a positive impact on the value of green brands, and the effect of the eastern region is significantly better than that of the central and western regions. (2) The spatial spillover effect of the two stages of regional innovation efficiency on the value of green brands is evident, especially in the eastern region. (3) The innovation value chain has a pronounced spillover effect. (4) The single threshold effect of intellectual property protection is significant. When the threshold is crossed, the positive impact of the two stages of innovation efficiency on the value of green brands is significantly enhanced. (5) The influence of economic development level, openness, market size, and marketization degree on the value of green brands shows remarkable regional differences. In conclusion, this study contributes to understanding green brands’ growth and provides important implications for developing independent brands in various regions of China.

## 1. Introduction

Reviewing the development process of global modernization, the rapid growth of the global economy has also created many ecological and environmental problems, such as climate change/global warming, increased pollution, and resource shortages. Since the 1990s, countries worldwide have paid more attention to green coordination and sustainable development of the economy and ecological environment. Nowadays, green development, a form of economic growth and social development aimed at efficiency, harmony, and sustainability, has become a significant trend globally. Many countries worldwide consider the development of green industries essential to promoting economic restructuring.

China has entered a new stage of high-quality economic development. Developing a green economy that can reduce damage to the ecological environment and achieve sustainable development is a critical aspect of high-quality economic development. With the overall green transformation of China’s economic and social development, people’s consumption concepts and structure have also begun to change. More and more attention has been paid to product safety, food health, quality of the living environment, and other issues. Green consumption, a collective term for various consumption behaviors and patterns that meet human health and environmental protection standards, has gradually become popular. The popularity of green consumption has promoted the rapid development of green markets that specialize in selling products that produce little environmental pollution during production and consumption. In order to expand the green market share, obtain differentiated competitive advantages, and establish good customer relations, the green brand strategy has become an inevitable choice for enterprises to adapt to the green consumption wave. More and more enterprises in China are seeking a green development path. Huawei released the Green Development 2030 report, pointing out that green development is the key to breaking the future of enterprises. BYD announced a “fuel cut-off”, becoming the first auto company in the world to officially stop production of fuel vehicles. HSBC actively promotes “paperless bank” and “green credit”. It can be seen that establishing the green brand image of enterprises, creating green brand innovation, and promoting green innovation are the mainstream trends of the future development of all kinds of enterprises. Building a green brand is an inevitable requirement in order for enterprises to enhance their competitiveness and ensure sustainable development. According to the Green Ranking released by Newsweek in 2017, 52 enterprises in China have entered the global top 500, 148 in the United States, 60 in Japan, and 32 in France. No Chinese enterprise has joined the international top 50, but there are 15 in the United States, 6 in France, and 3 in Japan. As the second-largest economy in the world, China has made some achievements in developing green brands. However, there still needs to be a gap in the quality and efficiency of green brand growth compared with the United States, Britain, Japan, and other countries. The green brand is not limited to the category of ecological and environmental protection. However, it is closely related to the sustainable development of enterprises. Developing green brands not only conforms to the development trend of the social environment, but also conforms to the wave of green consumption in the market. At the same time, it is also conducive to improving enterprises’ international competitiveness and sustainable development ability. As China’s green market is not mature enough, problems such as poor authenticity of green brands, “hollowing out” of green brands, and “green floating” of brands have begun to emerge, which have seriously affected consumers’ enthusiasm for green brand consumption and green brand trust [1]. How to create a green brand with consumer trust and value has become a hot issue in business and academic circles.

The current research on green brands is conducted chiefly from the perspective of consumers and enterprises [2,3,4,5], and research on green brand value based on a regional perspective is rare. Previous studies have shown that innovation capability can help enterprises gain competitiveness and sustainability and thus help enterprises improve their market position, establish a brand reputation, bypass competition, make breakthroughs, and attract customers [6,7,8,9]. However, previous studies only discussed whether innovation could improve brand value and generally regarded the innovation process as a “black box”, requiring more analysis of the innovation process. Only a few studies conducted independent research on green brands. In addition, the development of brands and the process of sci-tech innovation both need the protection of intellectual property rights as part of their premise. Legal and institutional means are required to protect enterprises’ innovation achievements and encourage enterprises to continue to innovate to inject fresh blood into the development of enterprise brands constantly.

Given the above background and existing research, this paper, from the perspective of the innovation value chain, divides the innovation process into two stages: R&D and achievement transformation, to more clearly reveal the impact of the two stages of innovation on green brand value. As the degree of intellectual property protection varies significantly in different regions of China, taking intellectual property protection as a threshold variable, we can explore the impact of two-stage innovation on green brand value under different levels of intellectual property protection. On the one hand, we can further reveal the differences in the impact of innovation at different stages on green brand value; on the other hand, we can provide targeted policy recommendations for developing green brands in different regions. As for the impact of regional innovation on brands, existing studies usually regard the research region as a whole, which ignores the spatial relevance of innovation activities and other economic activities among regions. The establishment of a spatial Dubin model can further discuss the impact of this spatial effect on green brand value. To provide a theoretical and practical basis for sci-tech innovation to promote the green brand value and provide targeted policy suggestions for constructing green brands in different regions of China.

## 2. Literature Review

### 2.1. Green Brand and Influencing Factors of Brand Value

Green brand refers to specific brand characteristics and attributes related to reducing environmental impact and consumers’ different environmental demands [2]. Compared with nongreen brands, green brands have three characteristics: greenness, sustainability, and externality. Greenness refers to green brands’ function of improving the ecological environment and social environment, which allows them to achieve a “win-win” in population, economy, environment, and other aspects. Sustainability refers to the efficient and reasonable allocation of enterprise resources due to the green nature of green brands. Externality refers to the positive impact of enterprises’ development of green brands on other economic entities, such as improving the ecological environment, guiding other enterprises towards green practices, leading consumers to green consumption, etc. Scholars have carried out a series of studies in green brand-related fields, mainly from the perspective of consumer behavior. Royne et al. (2011), Hartmann & Apaolaza-Ibáez (2012), Suki (2016), and other researchers found that the main factors that affect consumers’ green brand choice and purchase behavior are the deterioration of the external ecological environment and the enhancement of their health and environmental awareness [10,11,12]. Consumers have the motivation to choose green brands, but the actual efficiency of green brand selection is not high. From the consumers’ perspective, the brand–consumer distance is too large, which is why consumers reject green brands. Enterprises can encourage consumers to establish green brand memory through green marketing, shorten the brand–consumer distance, and thus promote green brand consumption [13,14]. From the perspective of enterprises, green brand innovation is not vital, and uneven product quality is also an essential factor that restricts consumers’ green consumption [15]. Therefore, enterprises should pay attention to the role of green marketing in adjusting brand environmental relevance and consumers’ green brand attitude and the driving role of sci-tech innovation on the brand to enhance the value of green brands through innovation [9].

Brand value is the amount obtained by calculating all brand assets with a method similar to tangible assets evaluation [16]. Such brand assets include the value added by the brand to product sales in the market, as well as the cognition, attitude, and behavior of consumers and other stakeholders towards the brand. In short, brand value is the total value of all brand assets, expressed in monetary terms. Brand value is the most intuitive embodiment of brand competitiveness and the most direct reflection of a brand’s position in the market, as well as its development and change. The mainstream evaluation methods for brand value include Interbrand, Financial World, World Brand Lab, etc. With the expansion of the influence of the World Brand Lab in China, Chinese scholars usually use the brand value data released by the World Brand Lab to conduct relevant empirical research [17,18,19].

Currently, the research on green brand value continues the research method of brand value and is mainly based on two perspectives of corporate finance and consumers. Chen (2010) believed that green brand value is a series of brand assets and liabilities related to enterprises’ green commitments and environmental concerns [20]. Ng et al. (2014) pointed out from the consumers’ perspective that green brand value is the overall evaluation of consumers’ perception of green products or services and their environmental desire, sustainable expectation, and green demand [21]. With the development of society, the evaluation of corporate brands, especially green brands, should not be limited to traditional financial indicators such as market value, operating income, profitability, or the green value perceived by consumers. Whether there are positive environmental and social externalities is also an important consideration. ESG index is an evaluation standard system for enterprise, which mainly encompasses three aspects: the impact of enterprise on the environment (E), responsibility to society (S), and internal governance (G). The three aspects are closely related to the three characteristics of the green brand (green, sustainability, and externality). The ESG index can reflect the development of green brands of enterprises to a certain extent. Therefore, this paper will measure the regional green brand value based on the brand value data released by the world brand experiment and the ESG index released by Shanghai Huazheng Index Information Service Co., Ltd.

### 2.2. Sci-Tech Innovation and Brand Value

The idea of sci-tech innovation as an essential source of brand value is relatively new, and issues of brand and sci-tech innovation penetration are gradually emerging in academia [22]. Aaker (1996) and Zhang et al. (2013) believe that sci-tech innovations, such as the inclusion of significant new technology attributes, may cause consumers to recognize convenience and comfort from the new technology attributes and appreciate innovation efforts and capabilities, thus creating a better image for the brand [23,24]. Kliestikova & Kovacova (2017) believe that innovation is being integrated into the construction and management of brands and use questionnaire surveys, choice analysis, and cluster analysis to empirically verify that innovation is an essential source of brand value perceived by consumers [25]. Kurt (2019) emphasized that companies focusing on R&D strategies to provide products based on technological innovation will contribute to brand value and corporate revenue in the global environment of immediate consumption, and empirical studies have shown a positive relationship between R&D expenditures, revenue, and brand value [26]. Yao et al. (2019) believe that sci-tech innovation mainly helps to improve production efficiency and product quality, thereby gaining long-term competitive advantages, which will be reflected in brand value. They also found that technical innovation has a stronger impact on improving brand value compared with nontechnical innovation [27]. Apparently, sci-tech innovation has an essential impact on brand value.

As for the relationship between sci-tech innovation and brand value, the existing research mainly carries out relevant research on two levels: firstly, at the enterprise level, based on the theory of enterprise resource base and the theory of core competitiveness; second, at the regional level, based on the theory of brand growth environment. Scholars generally argue that innovation can promote brand value by developing new products and services, improving the quality of existing products and services, increasing the added value of products and services, and other ways [28,29,30].

#### 2.2.1. The Perspective of Enterprise Resource Base and Core Competence

According to the theory of enterprise resource base, the resources owned by enterprises are the material basis for the construction and development of enterprise brands. High-quality enterprise resources can promote new brands’ success and help existing brands grow [28]. M’Zungu et al. (2010) pointed out that sufficient resources can guarantee enterprise’ R&D and production activities, which is conducive to the development and growth of enterprise brands [31]. Han & Zhao (2008) considered that for the development of China’s brands, improving product quality is the primary condition, and the quality of products depends not only on good product design but also on the skilled workers and high-quality knowledge of workers involved in enterprises [32]. Zhu & Wang (2018), based on the perspective of intellectual property rights, analyzed that enterprises’ intellectual assets, such as talents, patents, and trademarks, play an essential role in promoting brand competitiveness [33]. It can be seen that the innovation resources of enterprises will have a positive effect on brand competitiveness and brand value.

An enterprise’s core competence is a comprehensive system composed of the accumulation of knowledge, special skills, and related resources in the production and operation process of the enterprise. It is the power source of the sustainable competitiveness of the enterprise. Huang & He (2015) believed that brand competitiveness came from the core competence of enterprises, especially the ability to innovate independently [34]. The core competitiveness of an enterprise originates from its resources, but unique and difficult-to-imitate essential resources can bring long-term competitive advantages to the enterprise. The innovation resources of an enterprise are usually unique to that enterprise and are scarce, nonimitative, and irreplaceable [35]. Li & Liu (2017) believe that knowledge resources dominated by core technologies are the most important manifestation of enterprise innovation resources, which are the essential resources needed to cultivate the core competence of enterprises [30]. Through the Internet industry, Helm (2007) found that for the high-tech industry, product innovation and technology development make the brand different and promote the development of the industry [36]. Wang & Wang (2020) argued that the value of China’s time-honored brands comes from the accumulation of historical culture and the inheritance of core technologies, cultural endowment determines the direction of brand development, and the improvement and innovation of core technologies is the source of brand growth [37]. Based on enterprise resource theory and enterprise core competence theory, Wang et al. (2013), explored the influencing factors of independent brand creation and proposed that independent innovation is the fundamental means for Chinese enterprises to create high-quality brands while providing human capital quality is the key to enhance enterprise innovation ability [38].

To sum up, from the perspective of enterprise resources or core competence, improving enterprise innovation capability is essential to increase brand equity, improve brand competitiveness, achieve brand differentiation, and promote brand value.

#### 2.2.2. The Perspective of Brand Growth Environment

Brand building is affected not only by internal factors of the enterprise but also by external environmental factors of the enterprise; that is, the environmental factors of the region where the brand is located. Wang & Cheng (2012) built a unitary linear regression model and found that regional technological innovation has a significant positive impact on brand value without interfering with other influencing factors [39]. Wang et al. (2019) found that regional R&D capabilities and technological environment support significantly impact brand value by building a multiple regression model [40]. Zhou et al. (2014) calculated the regional technological innovation scores of 31 provinces by factor analysis and certified that the stronger the regional technological innovation ability, the higher the brand value [41]. Qi & Liu (2015) analyzed the impact of collaborative innovation and performance on the competitiveness of brands from the regional level and found that the higher the degree of collaboration between innovation subjects, the more vital the innovation efficiency, and thus the more significant the role of improving the competitiveness of brands [42]. In addition to the innovation environment, the regional market environment, legal system environment, social environment, political environment, and natural environment in which the brand is located will have a specific impact on the construction and growth of the brand [43,44].

Intellectual property protection is crucial in technological innovation and brand building. The more perfect the legal environment is, the higher the return rate of R&D investment and brand investment is, and the stronger the motivation of enterprises to innovate. Sukarmijan & Sapong (2014) believed that based on the background of the intellectual economy, intellectual property protection plays a vital role in promoting technological progress and brand promotion [45]. Wang et al. (2015) used panel data from 25 provinces in China to demonstrate that intellectual property protection can indirectly affect brand growth by influencing regional technological innovation capability [46]. Yan (2018) stressed that the government should strengthen the protection of agricultural products’ knowledge innovation, and create a good knowledge protection environment, to improve agricultural products’ brand value [47].

### 2.3. Innovation Value Chain Theory

Previous studies usually regarded the innovation process as a “black box”, which would ignore the internal structure and internal operating mechanism of innovation. From the perspective of the innovation value chain, we can explore the internal mechanism of the innovation process and match the research on the path of innovation efficiency improvement. The innovation value chain was first proposed by Hansen & Birkinshaw (2007). They argued that the innovation value chain could be divided into three stages: the generation, transformation, and dissemination of creativity, and there is a progressive internal correlation in the three stages [48]. Chinese scholars usually regard the innovation value chain as the decomposition of sci-tech innovation links based on the perspective of production. Sci-tech innovation is a multi-stage, multi-factor value chain transmission process from the input of innovative resources to the output of innovative products, which mainly includes the following three stages: the input of innovation, the condensation of innovative knowledge, and the realization of innovative achievements [49,50,51]. Yu and Liu (2014) divided the sci-tech innovation process into knowledge innovation, R&D innovation, and product innovation. They investigated the innovation efficiency of different provinces at different stages using the three-stage DEA model [49]. Considering that this paper focuses on analyzing the impact of technological innovation capability and value transformation capability on the development of green brands, the innovation process is simplified into two stages: R&D and achievement transformation. R&D is the basis of innovation, focusing on knowledge creation and technological research and development. Achievement transformation is the application of innovation to realize the economic value of innovation. Both stages of innovation have input–output functions and are interrelated processes. That is, the output in the R&D stage is usually the input in the achievement transformation stage. See Figure 1 for the two-stage innovation value chain model.

### 2.4. Dissipative Structure Theory

In addition to being widely used in physics, chemistry, and mathematics, dissipative structure theory has gradually become a new research paradigm in economics. Its solid economic explanatory power has laid a foundation for its extensive economic application. Perrings (1986) applied the theory of dissipative structure to the analysis of economic environment systems and pointed out that economic environment systems are complex systems with dissipative structure properties [52]. The dissipative structure has four characteristics: openness, nonequilibrium, internal nonlinearity of the system, and an internal driving effect of “fluctuation” [53,54]. The brand ecosystem has four main dissipative structure characteristics as a complex economic system. As a component of the economic system, the brand ecosystem has a close exchange of capital, technology, information, and other elements with other systems in the economic system (openness). The brand ecosystem is not static, but a nonequilibrium evolution system. Technological progress or changes in market demand bring about brand changes, with which the brand ecosystem is gradually adjusted away from the original equilibrium. With the maturity of new technologies or the stability of market demand, the brand ecosystem enters a new equilibrium (nonequilibrium). The development of any brand will be affected by other brands and stakeholders. There is a complex network of associations between brands and between brands and stakeholders that compete with and promote each other [55]. It is not a simple linear relationship that can be described and depicted (in other words, it is nonlinear). “Fluctuation” originates from the change in policy, capital, technology, market, stakeholders, and other influencing factors. Any small influencing factor deviating from the original equilibrium state will be further amplified into a “huge fluctuation” that controls the evolution of the whole system through the nonlinear interaction relationship (the internal driving role of “fluctuation”). Therefore, the brand ecosystem is a system with the characteristics of a dissipative structure.

### 2.5. Summary of This Chapter

Scholars have conducted extensive research on the relationship between innovation and brand and have gathered rich research results. Mainly based on the theory of brand equity and brand competitiveness, they believe that innovation can increase brand equity and improve brand competitiveness, thereby improving brand value and brand influence. Brand equity theory and brand competitiveness theory are the extension and expansion of resource-based theory and core competence theory in marketing. Essentially, they both emphasize that only continuous innovation can provide inexhaustible power for the development and growth of brands.

Nevertheless, scholars seldom explore the impact of innovation on green brand value from the regional level and usually regard the research region as an independent whole, which will ignore the differences between regional development and the spatial interaction of regional economic activities. Secondly, innovation is a multi-stage and multi-output process, and the impact of innovation output at different stages on brand value may differ. Thirdly, in addition to the innovation environment in which the brand is located, the market environment, legal environment, social environment, etc., will have a particular impact on the creation and growth of the brand; in particular, the legal environment of intellectual property. On the one hand, intellectual property protection can provide a legal basis and protection for enterprises to safeguard brand rights and create a good external environment and institutional guarantee for enterprises to promote brand growth. On the other hand, it can effectively weaken the externality of innovation and avoid the phenomenon of “free riding”, to protect the innovation achievements of enterprises, ensure the innovation benefits of enterprises, and improve the enthusiasm of enterprises for continuous innovation. Finally, according to the four characteristics of the dissipative structure, this paper discusses that the brand ecosystem is a system with the characteristics of the dissipative structure.

## 3. Study Design

### 3.1. Model Construction

According to the brand ecosystem theory, the brand ecosystem is a business ecosystem composed of brands and their related environments for survival and development, including government, market, sci-tech innovation, legal system, culture, and other ecological environment elements, investors, suppliers, industry associations, customers, and other relevant stakeholder elements [56]. The construction of a green brand is carried out under specific environments and conditions. It is closely related to sound economic development, a high level of opening to the outside world, a perfect market, a large market scale, and other factors. It is encouraged and constrained by various factors and relationships between all parties. Any change in these factors will affect the realization of green brand value. According to the dissipative structure theory of Prigogine (1994) [57], we can infer that if the brand ecosystem is an isolated system, then according to the principle of entropy increase, the entropy of the system will continue to increase, and the perfection of the system will certainly weaken. Suppose the brand ecosystem is an open system. In that case, the introduction of negative entropy from the surrounding environment will offset the increase in system entropy by constantly exchanging material, energy, and information with the outside world, giving the brand ecosystem the characteristics of a dissipative structure. This dissipative structure feature will make the brand ecosystem more orderly and promote green brand value in the system. Regional sci-tech innovation is an important way for the system to obtain the negative entropy flow, while protecting intellectual property rights will enhance the negative entropy flow.

For an open system with dissipative structure characteristics, its development state can be measured by calculating the total entropy change of the system [53]. The total entropy change (*dS*) of the system comes from the positive entropy flow (*dSi, dSi* > 0) generated by the system itself and the negative entropy flow (*dSe, dSe* < 0) formed by the system’s exchange with the outside world. The formula is *dS* = *dSi* + *dSe*. When *dS* < 0, the system will have a self-organization phenomenon and gradually evolve from the low stage to the high stage. When *dS* > 0, the system will be in a disordered state of change and degenerate from the advanced stage to the low-level stage. When *dS* = 0, it means that the system has not changed. Based on the theory of dissipative structure and the principle of entropy increase, this paper builds a relationship model between sci-tech innovation and the promotion of green brand value, as shown in Figure 2.

Sci-tech innovation ability is the source of power to promote the development of green brands. From the regional perspective, the value of a green brand in a region is not only affected by the local brand’s ecological and environmental factors, but also neighboring regions’ environmental factors. In addition, the strength of intellectual property protection will affect the role of technological innovation in promoting the value of green brands. According to the above analysis, the empirical test model constructed in this paper is shown in Figure 3.

#### 3.1.1. Two-Way Fixed Effects Model

In order to study the impact of two-stage innovation efficiency on the value of the green brand, this paper uses the stepwise regression method to build Models (1)–(3). The specific model forms are as follows:(1)$${BV}_{it}=\alpha_{0}+\alpha_{1}{TRD}_{it}+\alpha_{2}{EDL}_{it}+\alpha_{3}{OUL}_{it}+\alpha_{4}{MS}_{it}+\alpha_{5}{MD}_{it}+\mu_{it}$$
(2)$${BV}_{it}=\alpha_{0}+\alpha_{1}{TAT}_{it}+\alpha_{2}{EDL}_{it}+\alpha_{3}{OUL}_{it}+\alpha_{4}{MS}_{it}+\alpha_{5}{MD}_{it}+\mu_{it}$$
(3)$${BV}_{it}=\alpha_{0}+\alpha_{1}{TRD}_{it}+\alpha_{2}{TAT}_{it}+\alpha_{3}{EDL}_{it}+\alpha_{4}{OUL}_{it}+\alpha_{5}{MS}_{it}+\alpha_{6}{MD}_{it}+\mu_{it}$$

The explained variable *BV_it_* is the green brand value of region *i* in year *t*. The explanatory variables *TRD_it_* and *TAT_it_* are the R&D efficiency and achievement transformation efficiency of region *i* in year *t*; *EDL_it_*, *OUL_it_*, *MS_it_*, and *MD_it_*, respectively represent the regional economic development level, the degree of opening to the outside world, the market size and the degree of marketization. α is the regression coefficient; μ is a random error term.

#### 3.1.2. Two-Way Fixed Effects Model

According to the application of “the first law of geography” in economics, there is a specific interaction between regional economic activities in space [58]. Commonly used spatial econometric models include the spatial Doberman model (SDM), spatial lag model (SLM), and spatial error model (SEM). SDM has more real explanatory power than the other two models because it can examine the influence of explanatory variables in adjacent areas on the explained variables [59]. Therefore, to avoid ignoring the possible model estimation bias caused by spatial effects of regional economic behavior and to make the research results more realistic, this paper explores the spatial effects of regional innovation efficiency on the value of green brands by building a SDM. The specific manifestations of the SDM are as follows:(4)$$\begin{array}{l} BV_{it}=\rho WBV_{it}+\beta_{0}TRD_{it}+\beta_{1}EDL_{it}+\beta_{2}OUL_{it}+\beta_{3}MS_{it}+\beta_{4}MD_{it}\\ +\theta_{0}TRD_{it}+\theta_{1}EDL_{it}+\theta_{2}OUL_{it}+\theta_{3}MS_{it}+\theta_{4}MD_{it}+\varepsilon_{it} \end{array}$$
(5)$$\begin{array}{l} BV_{it}=\rho WBV_{it}+\beta_{0}TAT_{it}+\beta_{1}EDL_{it}+\beta_{2}OUL_{it}+\beta_{3}MS_{it}+\beta_{4}MD_{it}\\ +\theta_{0}TAT_{it}+\theta_{1}EDL_{it}+\theta_{2}OUL_{it}+\theta_{3}MS_{it}+\theta_{4}MD_{it}+\varepsilon_{it} \end{array}$$
(6)$$\begin{array}{l} BV_{it}=\rho WBV_{it}+\beta_{0}TRD_{it}+\beta_{1}TAT_{it}+\beta_{2}EDL_{it}+\beta_{3}OUL_{it}+\beta_{4}MS_{it}+\beta_{5}MD_{it}\\ +\theta_{0}TRD_{it}+\theta_{1}TAT_{it}+\theta_{2}EDL_{it}+\theta_{3}OUL_{it}+\theta_{4}MS_{it}+\theta_{5}MD_{it}+\varepsilon_{it} \end{array}$$

*W* is the spatial weight matrix; *WBV_it_* is the spatial lag item of the explained variable’s green brand value; *WTRD_it_*, *WTAT_it_*, *WEDL_it_*, *WOUL_it_*, *WMS_it_,* and *WMD_it_* are the spatial lag items of R&D efficiency, achievement transformation efficiency, and other control variables, respectively. ρ represents the spatial autocorrelation coefficient, *β* and *θ* represent the regression coefficient, and *ε* represents the error term.

The spatial adjacency weight matrix (*W*) is set according to whether provinces are adjacent geographically. If two regions are adjacent, the matrix element is set to 1. If two regions are not adjacent, the matrix element is set to 0. The spatial inverse distance matrix (*W**) is set according to the Euclidean distance (*d_ij_*) between the provincial capitals of each province and city and is used for the subsequent robustness test. The weight matrix elements of the spatial adjacency weight matrix (*W*) and spatial inverse distance matrix (*W^*^*) are defined as follows:(7)Wij=1,region i is adjacent to region j0,region i isn’t adjacent to region j,i≠jWij*=1dij,i≠j0,i=j

#### 3.1.3. Panel Threshold Model

This paper refers to Hansen (1999)’s panel data threshold model [60], selects intellectual property protection as the threshold variable of R&D efficiency and achievement transformation efficiency, and constructs the threshold regression model as follows:(8)$$\begin{array}{l} {BV}_{it}=\alpha_{0}+\alpha_{1}{TRD}_{it}I({IPR}_{it}\leq \gamma )+\alpha_{2}{TRD}_{it}I({IPR}_{it}\geq \gamma )+\alpha_{3}{EDL}_{it}\\ +\alpha_{4}{OUL}_{it}+\alpha_{5}{MS}_{it}+\alpha_{6}{MD}_{it}+\mu_{it} \end{array}$$
(9)$$\begin{array}{l} {BV}_{it}=\alpha_{0}+\alpha_{1}{TAT}_{it}I({IPR}_{it}\leq \gamma )+\alpha_{2}{TAT}_{it}I({IPR}_{it}\geq \gamma )+\alpha_{3}{EDL}_{it}\\ +\alpha_{4}{OUL}_{it}+\alpha_{5}{MS}_{it}+\alpha_{6}{MD}_{it}+\mu_{it} \end{array}$$

*IPR_it_* is the level of intellectual property protection in year *t* of region *i*, *γ* is the threshold value, *ε* is a random disturbance term, and *I* (·) is a threshold index function.

### 3.2. Variable Design

#### 3.2.1. Explained Variables

Previous studies mostly explored the impact of sci-tech innovation on enterprise brand value from the enterprise level. This paper focuses on the impact of technological innovation on the overall level of green brand value in the region from the regional level. It considers the poor availability of regional green brand value data. Therefore, this paper’s regional green brand value is the sum of the regional green brand value. The measurement method is as follows: first, we count the brands that are shortlisted in China’s 500 Most Valuable Brands. Secondly, according to the ESG index, we remove the enterprise brands rated below B. Finally, according to the brand value data released by the World Brand Lab, the total brand value of enterprises in a particular region with an ESG rating of B or above is estimated.

#### 3.2.2. Explanatory Variables

This paper refers to the super-efficient SBM proposed by Tone (2002) [61] and uses the efficiency value to measure the two-stage innovation level: R&D efficiency (*TRD*) and achievement transformation efficiency (*TAT*). In the stage of R&D, R&D personnel and R&D funds are usually used as input in the initial stage of innovation. Innovation output is generally about knowledge and technology, and its manifestations are patents, inventions, monographs, and scientific papers. In the R&D stage, R&D personnel and R&D expenditure are selected as the innovation input indicators in this stage. The number of patent applications and Chinese scientific and technological papers included in three major foreign retrieval tools (SCI, EI, CPI-S) are selected as the innovation output indicators to measure technology and knowledge. Of these, the total R&D expenditure of each region is calculated as R&D capital stock (based on 2009) concerning the perpetual inventory method proposed by Pittman (1983) [62], *K_t_* = (1 − *δ*) *K_t−1_* + *I_t_*, where *K_t_* is the R&D capital stock in *t* period. It is the R&D expenditure in period *t*; *K_t−1_* is the R&D capital stock of period *t − 1*; *δ* is the capital depreciation rate. The calculation of initial R&D capital stock and depreciation rate refers to the practice of Shan (2008); with the capital depreciation rate *δ* set as 15%, and initial R&D capital stock *K_0_* = *I_0_*/(*δ* + *e*), *e* is the average growth rate of R&D expenditure [63]. The number of patent applications and scientific and technological papers published, as the intermediate variables of the innovation value chain, are not only the innovation output indicators in the R&D stage, but also the input indicators in the achievement transformation stage. Enterprises also need to provide corresponding financial support when developing and utilizing innovative achievements. Therefore, new product development expenditure is selected to reflect the investment of innovation funds in the achievement transformation stage. The innovation achievements will eventually provide economic benefits to the enterprise, so the sales revenue of new products and the export revenue of new products are selected as the innovation output in the achievement transformation stage. See Table 1 for the two-stage evaluation indicators of regional innovation efficiency.

#### 3.2.3. Threshold Variables

In order to measure the level of intellectual property protection (*IPR*), the GP index method proposed by Ginarte & Park (1997) is a quantitative measurement method commonly used abroad to measure the level of intellectual property protection [64]. The GP index measures the level of protection from the legislative level of intellectual property protection. Due to the relatively imperfect legal systems of developing countries, the level of intellectual property protection measured by the GP index will be on the high side. Thence, Chinese scholars usually measure the actual level of intellectual property protection in China based on the research ideas of Han & Li (2005), taking into account the law legislation and enforcement level of intellectual property protection [65]. Hu et al. (2012) proposed a new method to objectively and comprehensively measure the level of intellectual property protection by the proportion of technology market transactions in local GDP [66]. This method does not need to trace the factors that affect intellectual property protection, which are difficult to measure. Given the measurability and objectivity of this method, this paper uses the proportion of technology market transactions in local GDP to measure the level of regional intellectual property protection.

#### 3.2.4. Control Variables

According to the theory of brand growth environment, brand development is affected not only by the enterprise’s internal factors, but also by environmental factors. Considering the significant differences in the level of economic development, openness, market size, and marketization in various regions may have a particular impact on the growth of the green brand. Therefore, these factors are introduced as control variables. The regional economic development level (*EDL*) is measured by per capita GDP. The regional opening up level (*OUL*) is measured by the proportion of total imports and exports in local GDP. The market size (*MS*) of the region is measured by the total resident population of the region. The degree of marketization (*MD*) is measured by the proportion of the government’s general public budget expenditure to the local GDP. If the proportion is high, the government has more intervention in the market, and the degree of marketization is low. See Table 2 for specific measurement indicators of each variable in this paper.

### 3.3. Sample Selection and Data Source

The research sample of this paper is 25 provinces in China. Due to the lack of brand value data in some years in Tibet, Qinghai, Gansu, and other regions, these regions are not within the scope of the study, considering the continuity of data. There are 12 provinces in the east: Beijing, Fujian, Guangdong, Hebei, Heilongjiang, Jiangsu, Jilin, Liaoning, Shandong, Shanghai, Tianjin, and Zhejiang. There are 7 provinces in the central region: Anhui, Henan, Hubei, Hunan, Jiangxi, Inner Mongolia, and Shanxi; There are 6 provinces in the western region: Chongqing, Guangxi, Guizhou, Shaanxi, Sichuan, and Yunnan.

In calculating the total value of brands of enterprises listed in various regions, green brand screening is based on the Huazheng ESG index rating. Considering that the innovation characteristics of enterprises in the hotel, catering, jewelry, and other industries are not obvious enough, the data of such enterprises are excluded. In addition, since the brand value released by the World Brand Experiment is calculated based on the relevant data of the previous year; that is, the brand value released in 2021 is the brand value in 2020, so the statistical year of the green brand value is one year ahead of schedule. The innovation data of each province and city come from the China Science and Technology Statistics Yearbook over the years, and other data come from the China Statistics Yearbook and the statistical yearbooks of each province and city. Some missing data are processed by interpolation. This paper uses panel data, and the research range is from 1 January 2009 to 31 December 2020.

## 4. Empirical Analysis Process and Results

### 4.1. Time Change Trend of Green Brand Value and Two-Stage Innovation Efficiency in Three Major Regions of China

Before the empirical analysis, the time trend of green brand value and two-stage innovation efficiency in the three regions over the years is statistically analyzed. Figure 4 shows the development trend of green brand value in three regions of China from 2009 to 2020. From the overall trend, the value of green brands in China is steadily rising, and the development trend of green brands is good. However, the value of green brands varies significantly among regions. The growth trend of green brand value in the eastern region is significantly higher than in the central and western regions. This is because the eastern region has a relatively high level of economic development and opening to the outside world and a good market environment and innovation environment, which can contribute to the development of enterprises and brand building in the region.

Figure 5 shows the trends of R&D efficiency and achievement transformation efficiency in China’s regions from 2009 to 2020. From the overall trend, the efficiency of R&D and the efficiency of achievements transformation in China are rising. The trend of achievement transformation efficiency is greater than that of R&D efficiency. This shows that with the progress of science and technology and the deepening of market-oriented reform, China has made specific achievements in R&D and achievement transformation. Regarding R&D efficiency, the eastern region showed a steady growth trend, the central region showed an apparent upward trend after 2017, and the western region showed a downward trend after 2017. In terms of the efficiency of achievements transformation, the three regions are subject to volatile changes, and the fluctuation range is extensive. To a certain extent, the economic benefits of innovation will be affected by various factors, such as the regional economic environment, market changes, and policy changes. These factors vary significantly among different regions, leading to a volatile trend in the efficiency of innovation achievements transformation.

### 4.2. Impact of Two-Stage Innovation Efficiency on Green Brand Value

#### 4.2.1. Time Series Stationarity Test

This paper uses Stata16.0 software to conduct statistical analysis on sample data. The correlation analysis results show that explanatory and control variables are significantly correlated with the explained variables at 1%, which preliminarily verifies the rationality of the empirical model construction in this paper. The factor independence test results show that each variable’s variance inflation coefficient (VIF) is less than five, and the mean VIF is less than two, indicating that there are no multiple collinearities between variables and that the variable indicators are suitable.

Considering that this paper uses panel data to avoid the “pseudo regression” phenomenon, it is necessary to conduct a unit root test on panel data and judge the stability of the data. In this paper, the LLC test (in the case of the same unit root) and the Fisher ADF test (in the case of different unit roots) are used to test the stability of data. Table 3 shows that *BV*, *EDL*, and *MS* are unbalanced sequences under the horizontal sequence. However, under the first-order difference sequence, all variables reject the assumption that “there is a unit root”. Hence, all first-order differences are stable, and each variable is at least cointegrated with first-order units I (1).

#### 4.2.2. Regression Results and Analysis of Two-Way Fixed Effect Model

Above all, this paper uses the Hausman test to judge whether to choose a fixed or random effect model. The test results show that all models pass the significance test at least at the 10% level, so we choose the fixed effect model. Secondly, the White and Wooldridge tests are used to test whether the sample data have heteroscedasticity and autocorrelation. The results show that there are heteroscedasticity and autocorrelation. Therefore, this paper uses a two-way fixed effect model to control the time variables, and the Driscoll Kraay standard is used for error estimation. See Table 4 for the specific regression results.

From the national level, the impact coefficients of R&D efficiency and achievement transformation efficiency on the value of green brands are 0.3693 and 0.3286, respectively. Both have passed the significance test at the level of 1%, indicating that both can significantly promote the growth of green brand value. The regression results of national model (3) show that when the two innovation efficiency scores are simultaneously used as explanatory variables for regression analysis, the impact coefficients of R&D efficiency and achievement transformation efficiency are 0.3209 and 0.3057, respectively. Both pass the significance test at the 1% level, indicating that the R&D efficiency and achievement transformation efficiency positively impact the value of green brands together, which is synergistic.

In national Model (3), the impact coefficient of the economic development level is −0.0873, which is significant at the 10% level. Under normal circumstances, the higher the economic development level of a region, the better the brand development in the region. The negative impact may be that China’s green brand development started late, and the development of green brands seriously lags behind economic construction. The impact coefficient of the degree of opening up is 0.2541, and the significance level test shows that the higher the degree of opening up, the better the development of green brands. Improving the level of opening up and expanding the international market can promote China’s green brands from “China” to “the world”. China’s green brand value and influence can continuously improve through competition and cooperation with well-known international brands. The market size and the degree of marketization have positive and negative effects on the value of green brands, respectively. However, they need to pass the significance test, which the vast territory of China and the significant differences in resource endowments and market environments between different regions may cause.

At the regional level, the efficiency of R&D and the efficiency of achievements transformation in the eastern region positively impact the value of the green brand. The impact of the efficiency of achievements transformation is more significant than the efficiency of R&D due to the relatively developed market in the eastern region. In the developed market environment, once the new products enterprises develop successfully enter the market, the economic benefits will be incredible, which is conducive to improving the brand value. The R&D efficiency and achievements transformation efficiency in central China have not significantly impacted the value of green brands in this region. The sci-tech innovation efficiency in central China needs to be further improved. Attention should be paid to strengthening the economic transformation of innovation achievements. The efficiency of R&D and the efficiency of achievement transformation in the western region positively affect the value of green brands. The impact of R&D efficiency is far more significant than the efficiency of achievement transformation. While improving the level of R&D, the western region should also pay attention to the economic benefits of sci-tech achievements transformation.

The economic development level of the eastern region will have a significant positive impact on the green brand value. In contrast, the economic development level of the central and western regions has yet to have a significant positive impact. The development of the economy and green brands in the eastern region is relatively balanced. Only the level of opening up in the eastern region has a significant positive impact on green brand value, while only the market size in the western region has a significant negative impact on green brand value. This shows that the green brand market in the western region is still mainly domestic, while the eastern region has opened the international market. The degree of marketization in the eastern region positively impacts the value of green brands because the public economy dominates China’s primary economic system. The eastern region is the first to realize reform and opening up in China. Government market policies and economic support are an essential part of developing enterprises and brand building in this region. The degree of marketization in the central and western regions harms the value of green brands, indicating that improving marketization is conducive to developing green brands in the region.

In summation, improving the efficiency of regional R&D and the efficiency of achievements transformation can promote the growth of green brand value. The regional innovation efficiencies of the two stages have obvious synergy. However, due to the evident differences in the brand growth environment between regions, there are significant regional differences in the impact degree and effect of regional innovation efficiency of the two stages. The regional economic development level, the degree of opening up, the market size, and the degree of marketization also have significant regional differences in the impact of green brand value in different regions. Therefore, to speed up the construction and development of China’s green brands, it is necessary to improve the R&D ability and innovation achievements transformation ability, adapt to local conditions, and formulate a targeted development path according to the regional brand growth environment.

### 4.3. Analysis of Spatial Spillover Effect of Regional Innovation Efficiency on Green Brand Value

Referring to Elhorst (2014)’s idea of spatial applicability test [67], before building the spatial Dubin model, we should first use Moran’s I index to test the global spatial correlation of three core variables, namely, green brand value, R&D efficiency, and achievement transformation efficiency in 25 provinces. Furthermore, we depict the spatial aggregation of regional economic activities from the overall regional space. Moran’s I index is calculated as follows:(10)$$\begin{array}{l} \mathrm{Moran}’sI=\frac{\sum_{i=1}^{n}\sum_{j=1}^{n}w_{ij}(X_{i}-\bar{X})(X_{j}-\bar{X)}}{S^{2}\sum_{i=1}^{n}\sum_{j=1}^{n}w_{ij}}\\ S^{2}=\sum_{i=1}^{n}\frac{{\left(X_{i}-\bar{X}\right)}^{2}}{n},\bar{X}=\frac{1}{n}\sum_{i=1}^{n}X_{i} \end{array}$$
where *X_i_* represents the observation value of the ith region, *w_ij_* represents the standardized spatial adjacency matrix, and Moran’s *I* index is [−1,1]. If its value is greater than zero, this indicates that the data are positively correlated in space. If its value is equal to zero, this indicates that the data are spatially random. If the value is greater than zero, this indicates that the data are spatially negatively correlated. Next, we carry out the LM and robust Robust LM test on the residuals of nonspatial econometric models to further judge whether a spatial econometric model should be established. Finally, the applicability of the Spatial Doberman Model (SDM) is tested. LR and Wald tests are used to determine whether the SDM needs to degenerate into SEM or SLM.

#### 4.3.1. Spatial Correlation Test Results

Based on the spatial adjacency matrix (W), this paper uses Moran’s I index to test the spatial autocorrelation of green brand value, R&D efficiency, and achievement transformation efficiency. The global Moran’s I index and test results of core variables in each province and city from 2009 to 2020 are shown in Table 5. From 2009 to 2020, the overall Moran’s I index of green brand value is significantly negative. Most of them are significant at the 1% level, indicating that there is a negative spatial correlation between the green brand values of each province and city. That is, the value of green brands between regions presents the characteristics of “high-low” adjacent or “low-high” adjacent clustering. Based on the above analysis of the development of green brands in various provinces in China, some provinces have a high value of green brands, while neighboring provinces have a low value of green brands. It is not difficult to determine that there is a “siphon effect” in the development of green brands in China. That is, regions with a high development of green brands will attract talent, capital, technology, and other production factors from nearby regions due to their better market environment and economical level to promote their own region’s brand development while inhibiting the brand development of neighboring regions. However, this negative correlation is gradually weakening with the development of science and technology and more convenient transportation in recent years.

The overall Moran’s I index of R&D efficiency and achievement transformation efficiency from 2009 to 2020 is significantly positive and at least passes the 10% significance test, indicating a “spatial diffusion” effect of regional innovation in China. There is a positive interaction between R&D efficiency and achievement transformation efficiency in each region. The core variables have spatial correlation, which preliminarily verifies the rationality of using the spatial econometric model to conduct empirical research.

#### 4.3.2. Regression Results and Analysis of Spatial Dubin Model

According to the LM test and robust Robust LM test results, the panel data model has spatial autocorrelation, and a spatial panel regression model should be built. According to the LR and Wald test results, the SDM model is a better choice. Finally, according to the results of the Hausman test and the comparison of the goodness of fit, this paper uses the spatial Dubin model with fixed time effect as the benchmark return model. Regarding Elhorst (2010)’s practice [68], this paper uses the partial differential method to estimate the direct effect, indirect effect (spatial spillover effect), and total effect of two-stage innovation efficiency and other relevant influencing factors on green brand value. The direct effect represents the impact of local innovation efficiency on local green brand value. The indirect effect indicates the impact of local innovation efficiency on neighboring regions. The total effect represents the overall impact of innovation efficiency on green brand value. See Table 6 for the regression results of the spatial Dubin model.

From the national level, the direct effects of R&D efficiency and achievement transformation efficiency on green brand value are significantly positive, indicating that improving R&D and achievement transformation efficiency will significantly promote local green brand value. It can be seen from the national Model (6) that when the two stages of innovation efficiency are simultaneously used as explanatory variables for regression analysis, the influence coefficients of the direct effect and the total effect of R&D efficiency and achievement transformation efficiency are significantly increased. This shows that R&D efficiency and achievement transformation efficiency can jointly promote the increase of green brand value. They have the “1 + 1 > 2” effect. That is, the innovation value chain has a spillover effect.

The indirect effect coefficient of R&D efficiency on green brand value is 0.1239, which fails to pass the significance test, indicating that the positive impact of R&D efficiency on adjacent regions is insignificant. The indirect effect coefficient of achievement transformation efficiency is 0.6369, which passes the significance test. It shows that a spatial spillover effect on the impact of achievement transformation efficiency on the value of the green brand. The achievement transformation efficiency in this region has a significant role in promoting the value of the green brand in neighboring regions. The overall impact of R&D efficiency and achievement transformation efficiency on green brand value is significantly positive, and the total effect of achievement transformation efficiency is greater than that of R&D efficiency.

The regression results of national Model (6) show that the direct effect coefficient of economic development level on the value of the green brand is significantly positive at the level of 1%, the indirect effect is significantly negative, and the total effect is significantly positive. This indicates that if the value of the green brand is higher in regions with high economic development, the “siphon effect” will be generated to inhibit the development of the green brand in adjacent regions. The direct and total effects of the degree of opening up and market size are significantly positive. In contrast, the indirect effects are not significant, indicating that improving the level of opening up and exaggerating the market size is only conducive to the promotion of the value of the local green brand. The direct and total effects of the degree of marketization are significantly negative. In contrast, the indirect effect is insignificant, indicating that improving the level of marketization is only conducive to developing local green brand value.

From the regional level, the direct, indirect, and total effects of the efficiency of R&D and achievement transformation in the eastern region are significantly positive, and the three effect coefficients of achievement transformation efficiency are more significant than R&D efficiency. This reveals that the two-stage innovation efficiency in economically developed regions will not only positively impact the development of the local green brand, but will also promote the development of the green brand in neighboring regions. The value of the green brand will be more affected by achievement transformation efficiency. The three effects of the achievement transformation efficiency in the central region are insignificant, indicating that the achievement transformation efficiency has not played a significant role in promoting the green brand value of the region and adjacent regions. The direct effect of R&D efficiency in the central region is not significant, but the indirect effect is significantly positive, indicating that there is a problem of uncoordinated development between sci-tech innovation and green brands in some provinces in the central region; that is, provinces with solid sci-tech innovation experience poor development of the green brand. For example, the efficiency of R&D in Anhui Province ranks seventh in the country, but the value of the province’s green brand only ranks seventeenth. Therefore, the central region should focus on improving the ability to transform innovation achievements, improving the ability to coordinate the development of innovation and brand, and building a brand with innovation. The direct effect and total effect of R&D efficiency in the western region are significantly positive, indicating that R&D efficiency in the western region has a significant role in promoting the improvement of green brand value. However, the indirect effect is not significant, which may be caused by the low mobility of innovation factors among regions due to the limited geographical environment of the transportation industry in the western region. The three effect coefficients of achievement transformation efficiency in the western region are insignificant, indicating that the western region needs to improve the achievement transformation ability to develop its brand. This relies on innovation.

In summation, we can draw the following conclusions: improving the efficiency of regional R&D and the efficiency of achievements transformation will promote the improvement of the green brand value as a whole, and the innovation value chain has noticeable spillover effects. Regional innovation efficiency’s spatial effect on the green brand’s value in the two stages has noticeable regional differences. The spatial spillover effect of R&D efficiency in eastern and central regions is significantly positive, while the spatial spillover effect of achievement transformation efficiency is significantly positive only in eastern regions.

### 4.4. Analysis of Threshold Effect of Intellectual Property Protection

#### 4.4.1. Threshold Inspection

For the threshold test of intellectual property protection, firstly, we estimate the number of regression model thresholds when there is one threshold, two thresholds, and three thresholds using STATA16.0 software in Model (8) and Model (9), respectively. The estimation results are shown in Table 7. It can be seen from Table 7 that the single threshold model with the explanatory variable of R&D efficiency and the single threshold model with the explanatory variable of achievement transformation efficiency has passed the 5% significance test, and it is preliminarily judged that there is only one threshold in Models (8) and (9).

Secondly, we verify the authenticity of the threshold with the help of the likelihood ratio function graph, as shown in Figure 6. When LR = 0, the corresponding threshold parameter is the threshold estimate value. The confidence interval of the threshold estimate value is the threshold parameter interval corresponding to the LR value being less than the critical value under a specific significance level (when the significance is 5%, the critical value is 7.35) [69]. It can be seen from Figure 6 that the threshold estimate value of intellectual property protection in Model (8) (explanatory variable is *TRD*) is 0.0109, and the threshold estimate of intellectual property protection in the Model (9) (the explanatory variable is *TAT*) is 0.0064. The single threshold values of the two models are within their corresponding confidence zones, indicating that the single threshold estimates of the two models are consistent with the valid threshold values. It can be determined that there is only a single threshold for intellectual property protection in Models (8) and (9).

Therefore, this paper selects a single threshold model to analyze the threshold effect of intellectual property protection. The threshold value of intellectual property protection in the Model (8) is 0.0109, and its 95% confidence interval is [0.0104, 0.0110]. The threshold value of intellectual property protection in the Model (9) is 0.0064, and its 95% confidence interval is [0.0062, 0.0065].

#### 4.4.2. Regression Results and Analysis of Threshold Model

When the explanatory variable is R&D efficiency, the threshold model regression results are shown in Table 8, Model (8). When the explanatory variable is achievement transformation efficiency, the threshold model regression results are shown in Table 8, Model (9). According to the regression results of Model (8), the indirect impact of intellectual property protection on the value of the green brand through the efficiency of R&D shows a positive single threshold feature: When the level of intellectual property protection in a region does not exceed the threshold (0.0109), the regression coefficient of the impact of R&D efficiency on the value of the green brand is 0.3687, and it passes the 1% significance level test. When the level of intellectual property protection in a region crosses this threshold, the regression coefficient of the impact of R&D efficiency on the value of green brand increases to 0.7220 and passes the 1% significance level test. Eighteen provinces, including Beijing, Tianjin, Shaanxi, Shanghai, Hubei, and Jilin, have crossed the threshold of intellectual property protection (0.0109) by 2020. Compared to 2009, only Beijing, Shanghai, and Tianjin have crossed the threshold. It can be found that intellectual property protection in China has developed rapidly and achieved remarkable results over the past 12 years. However, the level of intellectual property protection in some regions is still low. It is still necessary to further strengthen the protection of intellectual property rights.

It can be seen from the regression results of Model (9) that the indirect impact of intellectual property protection on the value of the green brand through the efficiency of achievement transformation is also characterized by a single positive threshold: When the level of intellectual property protection in a region does not exceed the threshold (0.0064), the regression coefficient of the impact of achievement transformation efficiency on the value of the green brand is 0.2607, and it passes the 1% significance level test. When the level of intellectual property protection in a region crosses this threshold, the regression coefficient of the impact of achievements transformation efficiency on the value of green brand increases to 0.5027 and passes the 1% significance level test. By 2020, 20 provinces, including Beijing, Tianjin, Shaanxi, Shanghai, Hubei, and Jilin, have crossed the threshold of intellectual property protection (0.0064). Compared to 2009, only six provinces, including Beijing, Shanghai, Tianjin, Liaoning, Shaanxi, and Heilongjiang, have crossed this threshold, indicating that China’s intellectual property protection has also made remarkable achievements in the field of innovation achievements transformation. Compared with the threshold value of intellectual property protection in the field of achievement transformation, its threshold value in the field of R&D is higher. The reason for this may be that the innovation output in the stage of R&D usually takes the form of intangible assets such as knowledge and technology. Such intellectual property infringement is relatively secret, so the requirements for relevant legal systems and law enforcement are high.

To sum up, intellectual property protection has a significant threshold effect in the stage of R&D and the stage of achievement transformation. When intellectual property protection crosses the corresponding threshold, it will significantly improve the efficiency of R&D and the efficiency of achievement transformation on the positive effect of green brand value. Furthermore, because of the different innovation elements in each stage, the threshold effects of intellectual property protection on innovation efficiency in the two stages are quite different.

### 4.5. Robustness Test

Standard robustness testing methods include changing data sources, replacing variables, variable lag, model replacement, selecting subsamples, grouping regression, etc. Since the explained and explanatory variables in this paper are difficult to replace, grouping regression has been conducted according to regional division. Therefore, this paper tests the robustness of the benchmark model and the panel threshold model by lagging the explanatory variables by one period and tests the robustness of the spatial Dubin model by replacing the spatial adjacency matrix (*W*) with the spatial inverse distance matrix (*W**).

The robustness test results show that R&D efficiency and achievement transformation efficiency lagging behind by one period have a significant positive impact on the green brand value on the overall level. The efficiency of R&D has a greater impact on the green brand value than the achievement transformation efficiency. At the national level, the direct, indirect, and total effects of the efficiency of R&D are significantly positive, while the direct and total effects of achievement transformation efficiency are significantly positive, but the indirect effects are not significant. The protection of intellectual property rights has a significant single threshold effect on the efficiency of R&D and the efficiency of achievements transformation that are lagging by one phase. The above conclusions are consistent with the empirical results of the experimental model. Therefore, the model settings are robust, and the results are reliable.

## 5. Conclusions and Suggestions

### 5.1. Research Conclusions

Firstly, this paper estimates the total value of the green brand in 25 provinces in China from 2009 to 2020 using the data released by the World Brand Lab. The regional R&D efficiency and achievement transformation efficiency of each province and city in 12 years are calculated by building the super-efficiency SBM model. The paper provides a descriptive statistical analysis of the time and space change trend of the green brand value and the two-stage regional innovation efficiency of each province and city over a 12-year period, according to the calculation and measurement results. Secondly, the paper explores the impact of two-stage regional innovation efficiency and other control variables on green brand value by building a two-way fixed effect model as the benchmark model. With the help of SDM, this paper analyzes the spatial spillover effects of the two stages of regional innovation efficiency and other influencing factors. The level of intellectual property protection is introduced as a threshold variable to study the difference in the impacts of R&D efficiency and achievement transformation efficiency on green brand value under different levels of intellectual property protection. Finally, the main conclusions of this paper are as follows:

(1) The development trend of the green brand in various regions of China is good, but the differences among regions are large. The development of the green brand in the eastern region is significantly better than that in the central and western regions. The upward trend of green brand value is also significantly greater than that in the central and western regions, which indicates that the development of the green brand is better in regions with better market economic conditions and policy systems. In some regions, there is a “siphon effect” when developing the green brand. That is, regions with high brand development levels usually absorb brand resources and brand elements from surrounding regions, resulting in faster development in regions with a higher level of green brand development during slower development in regions with a low level of green brand development.

(2) The imbalance of innovation capability among regions in China is prominent, and there are differences among regions in different innovation stages. The eastern region is higher than the central and western regions in the efficiency of R&D and the efficiency of achievements transformation. This is due to the more developed economy and an advantage in the eastern region’s input and output of sci-tech innovation. The efficiency of R&D in the central region is lower than that in the western region, but the efficiency of achievements transformation is higher than that in the western region. The central region must strengthen institutional innovation and improve management efficiency. Due to its limited market development potential, the western region will inhibit the enthusiasm of enterprises to develop new products and expand the scale of market operation. Therefore, it is necessary to speed up the construction of the market system to stimulate innovation with demand. From the perspective of time trends, the efficiency of R&D and the efficiency of achievement transformation in the eastern and central regions are rising. However, the efficiency of R&D and achievement transformation in the western regions have declined in recent years. This is because the brain drain in the western regions has been severe in recent years, and a large number of outstanding scientific researchers, professors, scholars, etc., are flowing to the eastern and central regions, making the innovation ability of the western regions increasingly weak.

(3) The efficiency of R&D and the efficiency of achievements transformation will significantly enhance the value of the green brand. The efficiency of R&D and the efficiency of achievements transformation will have a significant value chain spillover effect. However, at the regional level, the two-stage innovation efficiencies of the central region have no significant impact on the value of the green brand. Enterprises in the central region should emphasize transforming innovation into productivity and improving the innovative value of products and services. There are regional differences in the impact of economic development level, the degree of opening up, market size, and marketization on the value of the green brand. The level of economic development in the eastern region has a significant positive impact on the value of the green brand. In the central region, there is a phenomenon that the development of brands needs to catch up with economic development. This may be because the innovation ability of the central region needs to promote brand development. The western region has yet to have a significant positive impact on the value of green brands due to its low level of economic development. Only the degree of opening up in the eastern region has a significant positive impact on the value of the green brand, and only the market size in the western region has a significant positive impact. It can be found that brand development in economically developed regions has shifted from market scale orientation to market scope orientation. The higher the level of marketization in the central and western regions, the greater the value of the green brand. The situation in the east is the opposite. Considering that this article measures the level of marketization by the degree of government intervention, it shows that the eastern region is a demonstration area for reform in many fields, which will promote the development and growth of the green brand to a certain extent.

(4) There is a significant single threshold effect in the level of intellectual property protection. When the level of intellectual property protection crosses the corresponding threshold, the positive effects of two-stage innovation efficiency on the value of an green brand are significantly improved. It can be seen that speeding up the construction of intellectual property laws and regulations, improving the judicial system, and strengthening law enforcement are of great significance to the development of China’s green brands.

(5) The two stages of regional innovation efficiency have spatial spillover effects on the value of the green brand, but there are significant regional differences. The spatial spillover effect of regional innovation efficiency in the two stages of the eastern region is significantly positive. In contrast, only the R&D efficiency in the central region has a significant positive spatial spillover effect. In comparison, the spatial spillover effect of the two stages of regional innovation efficiency in the western region is insignificant. In addition, there is a significant negative spatial spillover effect at the national level and in the eastern and central regions. That is, regions with a high level of economic development will harm the value of the green brand in their adjacent regions, which also explains why the development of the green brand in China has a “siphon effect”. This means that economically developed regions have better material living conditions, a market environment, policy support, and greater space for improvement, which will attract excellent talents, high-quality resources, and excellent brands from the surrounding regions, thus promoting the development of the local green brand. However, at the same time, this will have a certain inhibitory effect on the construction of green brands in the surrounding regions. The degree of opening up only produces positive spillover effects in the eastern and central regions. The market scale only produces a positive spillover effect in the western region. The degree of marketization has a positive spillover effect in the eastern region and a negative spillover effect in the western region, indicating that the policy support in the eastern region will radiate to the surrounding regions. In contrast, the marketization development in the western region will have a “crowding out effect” on the surrounding regions. In summary, the main reason for the existence of the “siphon effect” in China’s green brand development is that the “radiation effect” of regional sci-tech innovation and other factors is far less significant than the “siphon effect” of economic development.

### 5.2. Policy Suggestion

The brand economy is the product of the development of the market economy to a particular stage and is also a high-level manifestation of regional economic development. In the international market, the brand is not only a symbol of an enterprise but also a symbol of the competitiveness of a region or even a country. Improving the value and influence of green brands is of great practical significance to China’s green development. Therefore, based on the above research conclusions, this paper proposes the following suggestions:(1)Improve the two stages of innovation efficiency and emphasize the effect of sci-tech innovation in promoting brand value. While continuing to increase R&D funds, personnel, and other innovative resources, all localities should also pay attention to improving innovation output capacity, optimizing the allocation of innovation resources, and improving innovation efficiency, so as to provide strong power for China’s green brand building. The eastern region continues to produce a marked effect on R&D and achievement innovation in promoting the value of the green brand. The central region needs to strengthen institutional innovation, improve management efficiency, and formulate relevant policies to encourage enterprises to focus on products and quality, so as to produce a driving effect in innovation on brands. The western region needs to strengthen the construction of the market system, create a good market environment, shift the market competition from price and scale competition to product and service competition, and improve the corporate image and brand value with high-quality products and services. All regions should strengthen the degree of opening up to the outside world, improve marketization, and improve the construction of market mechanisms. Especially in the central and western regions, it should open up the international market, participate in competition and cooperation in the international market, take its essence, eliminate its dregs, and create a greener brand with international influence.(2)According to each region’s sci-tech innovation resource endowment, focus on superior resources and create a regional solid green brand. For example, the Jilin Province should encourage the development of green automobile brands (FAW, Hongqi, Jiefang, etc.); the Guangdong Province ought to promote the greening of household appliance brands (Gree, Midea, Skyworth, etc.); and the Jiangsu Province should vigorously develop green machinery manufacturing brands (Xugong, Hengtong Optoelectronics, Tongding, etc.).(3)Improve intellectual property laws and regulations and strengthen law enforcement and justice. The government should speed up the construction of laws and regulations system for intellectual property protection, implement the intellectual property protection law, strengthen the enforcement of intellectual property protection, and ensure the judicial fairness of intellectual property protection. Regional differences should be fully considered in policy formulation. Investment in sci-tech innovation should be increased for regions with a high level of intellectual property protection to promote the process of sci-tech innovation while improving the formulation of intellectual property protection for regions with a low level of intellectual property protection.(4)Give full play to the spatial spillover effect of R&D and achievements transformation, and strengthen regional innovation cooperation and communication, including the communication of scientific researchers, technologies, patents, management systems, and other innovative resources. The central government should overall construct a coordinated regional development mechanism, and its planning and requirements for regional economic development should not be limited to the local region. At the same time, it should consider its contribution to the coordinated development in regions and achieve win–win or multi-win through market mechanisms and benefit compensation mechanisms. The eastern region should support the central and western regions with redundant innovation resources. On the one hand, it can improve the innovation efficiency of the eastern region, and on the other hand, it can promote the innovation ability of the central and western regions. The central region should strengthen the spillover effect of innovation achievements and pay attention to the communication of innovation achievement transformation ability between regions. The western region needs to speed up the construction of transportation infrastructures such as expressways and high-speed rail, promote the construction of the market system and mechanism, drive innovation with demand, and promote green brand development with innovation. In areas with low economic development levels, the government should formulate relevant policies to improve the treatment of talents and strengthen support for enterprises so as to prevent brain drain and outflow of enterprise resources. Areas with high economic development levels should provide counterpart support to areas with low economic development, strengthen the interaction between universities, enterprises, and governments in their regions, and achieve win–win cooperation between regions in areas with high economic development levels.(5)Strengthen the spillover effect of the innovation value chain. In the market environment, the innovation competition is not only the competition for knowledge, technology, and other R&D capabilities but also the competition for research and development of new products, new markets, and other achievements and transformation capabilities. Therefore, all regions should promote the deep integration of IUR, improve the communication channels at all stages of the innovation value chain, and strengthen the effective interaction between the government, enterprises, universities, and research institutions, so that R&D and achievements transformation can form a benign interaction, producing “1 + 1 > 2” spillover effect.

### 5.3. Research Limitations and Future Research Directions

First of all, the sample selection could be better. Because of the availability and integrity of the data, this paper only studies 25 provinces in China. It excludes the sample data of nine provinces, such as Gansu, Xinjiang, and Tibet. Secondly, due to the need for more systematic and perfect evaluation methods for a regional green brand value in China, only the relevant data released by the World Brand Lab and Shanghai Huazheng are used. Although relatively authoritative, these data need more comprehensiveness and can only reflect a single region’s total green brand value. In contrast, each region’s green brand characteristics, influence, and structure are not considered. Finally, although this paper has divided three regions, including the eastern, middle, and western regions, there are still some limitations in the study areas. Regional development has presented the characteristics of urban clustering in recent years. The trend of urban clustering development in “Beijing-Tianjin-Hebei”, “YangtzeRiver Delta”, “Pearl River Delta”, and other cities is noticeable, and there are their regional development advantages. It impacts the region’s green brand development strategy and direction, so the regions need to be further divided.

With the rapid development of China’s market economy, developing the green brand economy is of great practical significance to realize the green transformation of economic development, improve China’s sustainable development ability and enhance its international competitive position. In the future, scholars can conduct more extensive and in-depth research in the following aspects: study various main elements of innovation (enterprises, universities, research institutions, government, etc.), nonmain elements (material conditions required for innovation), and coordinate the impact of policies and systems that various elements on green brand development from the static (innovation environment) or dynamic (innovation system) perspective. Improve the evaluation system of regional green brand development. Explore the path of brand innovation and development of small and medium-sized enterprises. Innovation drives green brand development and provides a relevant theoretical and practical basis for realizing the development of the green economy.

## Figures and Tables

**Figure 1 entropy-25-00290-f001:**
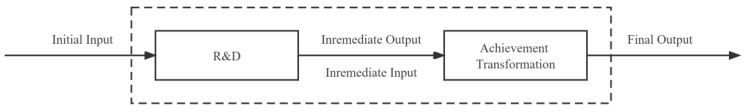
Two-stage innovation value chain.

**Figure 2 entropy-25-00290-f002:**
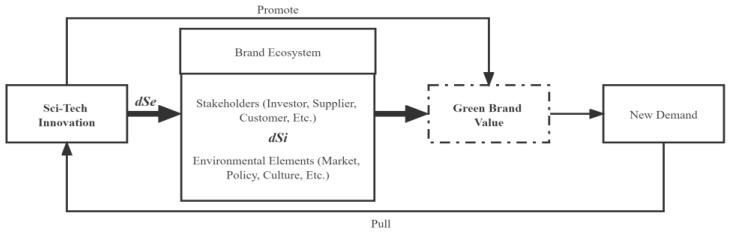
The relationship between sci-tech innovation and the promotion of green brand value.

**Figure 3 entropy-25-00290-f003:**
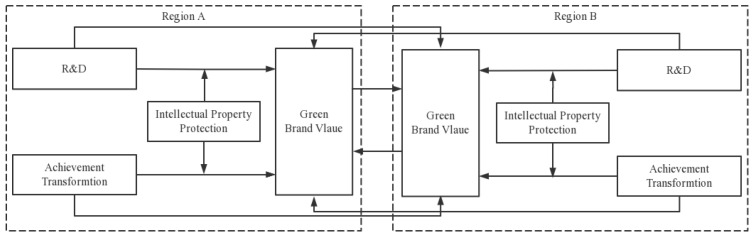
Influence mechanism of sci-tech innovation on green brand value.

**Figure 4 entropy-25-00290-f004:**
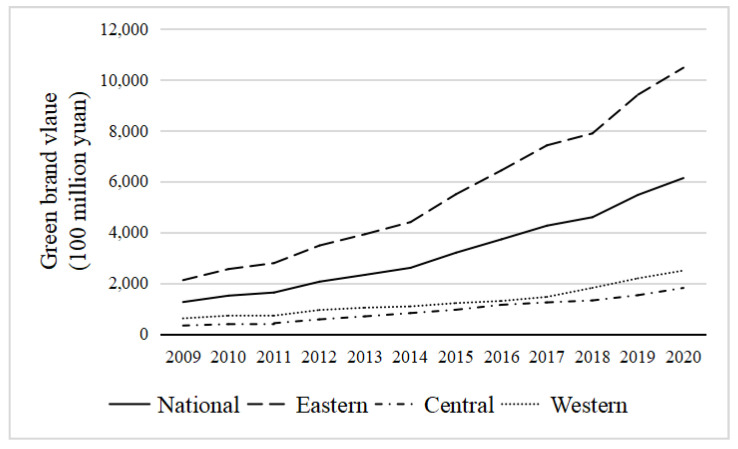
Change of average value of green brands in different regions of China from 2009 to 2020.

**Figure 5 entropy-25-00290-f005:**
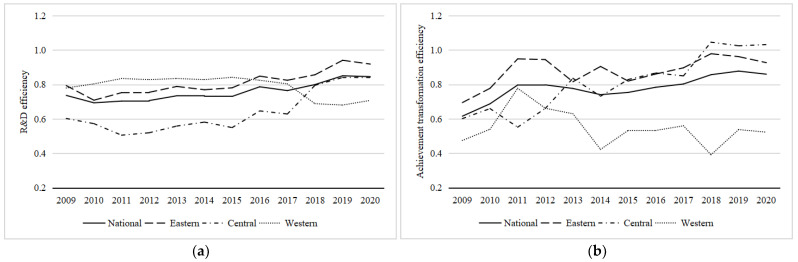
Changes in average R&D and achievement transformation efficiency in different regions of China from 2009 to 2020. (**a**) Change in R&D efficiency. (**b**) Change in achievement transformation.

**Figure 6 entropy-25-00290-f006:**
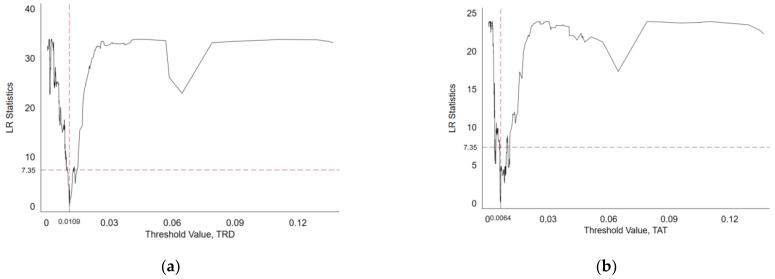
IPR threshold likelihood ratio function with TRD and TAT as explanatory variables. (**a**) IPR threshold likelihood ratio function with TRD as explanatory variable. (**b**) IPR threshold likelihood ratio function with TAT as explanatory variable.

**Table 1 entropy-25-00290-t001:** Two-stage regional innovation efficiency evaluation index system.

Stage	Indicator Type	Indicator Name
R&D	Input	R&D expenditure
		Full time equivalent of R&D personnel
	Output	Number of patent applications
		Number of scientific papers published
Achievements transformation	Input	Number of patent applications
		Number of scientific papers published
		New product development expenditure
	Output	Sales revenue of new products
		Export income of new products

**Table 2 entropy-25-00290-t002:** Variable Description.

Variable Type	Index Name	Indicator Measurement
Interpreted variable	Green Brand Value (*BV*)	The total value of all brands in China’s 500 Most Valuable Brands by region, taking the natural logarithm
Explanatory variable	R&D efficiency (*TRD*)	Calculated by super efficiency SBM model.
	Achievement transformation efficiency (*TAT*)	Calculated by super efficiency SBM model.
Threshold variable	Intellectual property protection level (*IPR*)	Technology market turnover divided by regional GDP
Control variable	Economic Development Level (*EDL*)	Per capita GDP of each region, taking natural logarithm
	Openness to the outside world (*OUL*)	Import and export volume divided by regional GDP
	Market size (*MS*)	The total number of permanent residents in each region, taking the natural logarithm
	Marketization degree (*MD*)	General public budget expenditure divided by regional GDP

**Table 3 entropy-25-00290-t003:** Descriptive statistics and correlation analysis.

Variable	LLC Statistics		ADF Statistics		Test Result (If the Sequence is Stable)
	Statistics	*p* Value	Statistics	*p* Value	
*BV*	−11.4794 ***	0.0000	25.2830	0.9986	No
*TRD*	−16.3135 ***	0.0000	105.9524 ***	0.0000	Yes
*TAT*	−9.6603 ***	0.0000	183.8841 ***	0.0000	Yes
*EDL*	−11.7807 ***	0.0000	46.2625	0.6241	No
*OUL*	−14.2522 ***	0.0000	73.6434 **	0.0164	Yes
*MS*	−7.9454	0.9102	191.2413 ***	0.0000	No
*MD*	−10.2202 ***	0.0000	132.1372 ***	0.0000	Yes
*L.BV*	−14.2425 ***	0.0000	67.7333 **	0.0481	Yes
*L.TRD*	−14.9561 ***	0.0000	102.6958 ***	0.0000	Yes
*L.TAT*	−8.7227 ***	0.0003	160.5342 ***	0.0000	Yes
*L.EDL*	−20.1709 ***	0.0000	71.3872 **	0.0252	Yes
*L.OUL*	−14.7298 ***	0.0000	66.0949 *	0.0632	Yes
*L.MS*	−18.3309 ***	0.0000	153.4215 ***	0.0000	Yes
*L.MD*	−9.1084 ***	0.0002	118.0936 ***	0.0000	Yes

Note: *, **, and *** are significant at 10%, 5%, and 1% statistical levels, respectively.

**Table 4 entropy-25-00290-t004:** Analysis of the impact of two-stage regional innovation efficiency on the value of green brands.

Variable	National			Eastern	Central	Western
	(1)	(2)	(3)	(3)	(3)	(3)
*TRD*	0.3696 ***		0.3209 ***	0.2036 **	−0.2288	0.9135 **
*TAT*		0.3286 ***	0.3057 ***	0.2500 ***	−0.0074	0.3188 **
*EDL*	−0.3564 *	−0.1698 *	−0.0873 *	0.9116 ***	−0.9127**	0.0268
*OUL*	0.3490 *	0.2055 *	0.2541 *	0.9955 ***	−0.3262	−0.9218
*MS*	0.4510	0.3007	0.6407	0.5371	0.9132	0.9811 ***
*MD*	−0.6199 **	−0.2212	−0.2017	0.7015 ***	−0.9166 **	−0.9217 **
C	5.9911 ***	5.3524 *	1.4011	−10.7175 ***	7.6209	−88.0073 ***
Time item	control	control	control	control	control	control
Sample size	300	300	300	144	84	72
R^2^	0.7395	0.7474	0.7552	0.8805	0.9311	0.6478
Hausman test	17.02 ***	16.19 **	14.86 **	15.62 **	52.25 ***	9.36 *
Model	FE	FE	FE	FE	FE	FE

Note: *, **, and *** are significant at 10%, 5%, and 1% statistical levels, respectively.

**Table 5 entropy-25-00290-t005:** The overall Moran’s *I* test results of green brand value and two-stage regional innovation efficiency.

Year		2010	2012	2014	2016	2018	2020
*BV*	Moran’s I	−0.211 ***	−0.203 ***	−0.208 ***	−0.202 **	−0.198 **	−0.164 **
	Z-statistic	−2.810	−2.919	−2.707	−2.283	−2.256	−1.680
	*p* value	0.002	0.002	0.003	0.011	0.012	0.046
*TRD*	Moran’s I	0.288 ***	0.146 *	0.165 *	0.189 **	0.223 **	0.189 **
	Z-statistic	2.509	1.443	1.591	1.909	2.197	1.945
	*p* value	0.006	0.075	0.056	0.028	0.014	0.026
*TAT*	Moran’s I	0.143 *	0.167 **	0.181 **	0.241 ***	0.296 ***	0.288 ***
	Z-statistic	1.535	1.720	2.418	3.084	2.802	2.737
	*p* value	0.062	0.043	0.018	0.001	0.003	0.003

Note: *, **, and *** are significant at 10%, 5%, and 1% statistical levels, respectively.

**Table 6 entropy-25-00290-t006:** Analysis of the spatial effect of two-stage regional innovation efficiency on the value of green brands.

	Variable	National			Eastern	Central	Western
		(4)	(5)	(6)	(6)	(6)	(6)
Direct effect	*TRD*	0.6025 ***		0.7767 ***	0.9179 ***	0.2497	0.5073 ***
	*TAT*		0.3111 **	0.3519 **	0.2563 **	0.0826	0.6742
	*EDL*	0.9144 ***	0.9188 ***	0.9197 ***	0.6074	0.9362 ***	−0.3645
	*OUL*	0.9123 ***	0.9104 ***	0.8205 **	0.1946	−0.5140	0.5933
	*MS*	0.8599 ***	0.9263 ***	0.9038 ***	0.8769 ***	0.8849 ***	0.9228 ***
	*MD*	−0.9110 **	−0.4710	−0.9842 **	0.9485 ***	0.9192 **	−0.9593 ***
Indirect effect	*TRD*	0.1239		0.0041	0.2135 *	0.7754 **	0.4614
	*TAT*		0.6369 ***	0.8021 ***	0.2543 **	0.0441	0.9184
	*EDL*	−0.9187 ***	−0.9191 ***	−0.9154 ***	−0.9107 **	−0.9255 ***	0.9261
	*OUL*	0.3715	0.4995	0.1159	0.9118 ***	0.9106 ***	0.9413 ***
	*MS*	−0.9152 ***	−0.9118 ***	−0.9104 ***	0.1772	0.1944	0.9490 ***
	*MD*	−0.9255 ***	−0.9126	−0.4261	−0.9573 ***	0.5581	0.9495 ***
Total effect	*TRD*	0.7275 ***		0.7809 ***	0.4446 ***	0.5258 **	0.9687 **
	*TAT*		0.7480 ***	0.9540 ***	0.5106 ***	0.1267	0.9117
	*EDL*	−0.4238	−0.0356	0.4269	−0.4617 ***	0.9111 **	0.9225 *
	*OUL*	0.9160 ***	0.9155 ***	0.9364 **	0.9137 ***	0.9101 ***	0.9100 ***
	*MS*	−0.6582 ***	−0.2545	−0.1319	0.9105 ***	0.9108 ***	0.9718 ***
	*MD*	−0.9365 ***	−0.9173 **	−0.9141 *	−0.8821	0.9248 ***	0.9492
	Sample size	300	300	300	144	84	72
	R^2^	0.7041	0.6876	0.7160	0.9366	0.7933	0.7538

Note: *, **, and *** are significant at 10%, 5%, and 1% statistical levels, respectively.

**Table 7 entropy-25-00290-t007:** Threshold effect test and estimation results of intellectual property protection.

	Threshold Effect Test				Threshold Estimation Results		
Explanatory Variable	Model	F Value	*p* Value	BS Times	Threshold	Estimated Value	95% Confidence Interval
*TRD*	Single threshold	34.57 **	0.0200	500	Single threshold	0.0109	[0.0104, 0.0110]
	Double threshold	12.09	0.1940	500	Double threshold	0.0017	[0.0015, 0.0018]
	Triple threshold	6.00	0.7400	500	Triple threshold	0.0036	[0.0035, 0.0037]
Conclusion	There is a single threshold						
*TAT*	Single threshold	25.42 **	0.0480	500	Single threshold	0.0064	[0.0062, 0.0065]
	Double threshold	5.90	0.4740	500	Double threshold	0.0104	[0.0102, 0.0106]
	Triple threshold	5.24	0.7180	500	Triple threshold	0.0016	[0.0015, 0.0016]
Conclusion	There is a single threshold						

Note: *, **, and *** are significant at 10%, 5%, and 1% statistical levels, respectively.

**Table 8 entropy-25-00290-t008:** Analysis of threshold effect of intellectual property protection.

Variable	(8)	(9)
*TRD* × I (*IPR* < 0.0109)	0.3686 ***	
*TRD* × I (*IPR* ≥ 0.0109)	0.7220 ***	
*TAT* × I (*IPR* < 0.0064)		0.2607 ***
*TAT* × I (*IPR* ≥ 0.0064)		0.5027 ***
*EDL*	−0.4106	−0.1166
*OUL*	0.3743 *	0.2364 *
*MS*	0.9844	0.2316
*MD*	−0.7061 *	−0.2550
C	2.0457	5.4149
Time item	control	control
Sample size	300	300
F statistic	77.22 ***	86.02 ***
R^2^	0.7602	0.7622

Note: *, **, and *** are significant at 10%, 5%, and 1% statistical levels, respectively.

## Data Availability

Not applicable.

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
