# Peer review of "Can the Sci-Tech Innovation Increase the China’s Green Brands Value?—Evidence from Threshold Effect and Spatial Dubin Model"

_entropy, 2023, doi:10.3390/e25020290_

Round 1
Reviewer 1 Report
Can the Sci-Tech Innovation Increase the China’s Green Brands Value?——Evidence from Threshold Effect and Spatial Dubin Model
Based on the perspective of the innovation value chain, sci-tech innovation is divided into two stages: R&D and achievement transformation. This paper uses panel data from 25 provinces in China as the sample. We utilize a two-way fixed effect model, spatial Dubin model, and panel threshold model to discuss the impact of two-stage innovation efficiency on the value of the green brand, the spatial effect of this impact, and the threshold role of intellectual property protection in the process. This article claims, among other things, that the two stages of innovation efficiency have a positive impact on the value of green brands.
I very much enjoyed reading this paper. The authors have done a great effort in writing it clearly and in the modern lenguaje and concepts of complexity and nonlinearity. I enjoyed reading their interpretation of economic phenomena as dissipative structues and the dynamics of fluctuations and amplification.
I recommend its publication. Minor comments:
1) Authors must define already in the introduction what they mean by green development, green economy, green consumption, green market, etc.
2) line 112 .. Etc must be etc.
Reviewer 2 Report
The article is a significant study of a significant amount of material and contains interesting results.
At the same time, the review of the literature needs minor refinement, there is a significant bias in the study of the literature from China. I would suggest supplementing the literature review with publications related to the impact of scientific and technical innovations on increasing the value of brands.
Some remarks are too radical (This paper first calculates) (P.22), (and the lack of previous research) (p,2) and are the subjective opinion of the authors.
The article offers a huge number of proposals, but not all of them are based on the analyzed data. However, there is a lack of specification of the purpose of the investigation, it is rather blurred in the article.
